# Assessing adherence to physical activity guidelines and correlates among older Korean adults with a focus on 10-minute bout duration using subjective and objective measures

Junhui Park[1], Su Hyun Kim[2]ᴥ*, Young Hoon Kim[3]ᴥ*, Chang-Hyung Lee[4]

**1** Department of Statistics and Data Science, Korea National Open University, Seoul, South Korea, **2** Industry-University Cooperation Foundation, Pukyong National University, Busan, South Korea, **3** Major of Marine Sports, Division of Smart Healthcare, College of Information Technology and Convergence, Pukyong National University, Busan, South Korea, **4** Department of Physical Medicine and Rehabilitation, Research Institute for Convergence of Biomedical Science and Technology, Pusan National University Yangsan Hospital, Pusan National University School of Medicine, Busan, South Korea

ᴥ These authors contributed equally to this work.
* shkim21@pknu.ac.kr (SHK); rehabkyh@pknu.ac.kr (YHK)

## Abstract

### Background

Understanding physical activity (PA) patterns in older adults is crucial for effectively promoting adherence to PA guidelines. However, measuring PA can be challenging because it involves a balance between ease of administration and accuracy of data collection. The primary objective of this study was to analyze PA levels in older Korean adults using both self-reported and objective accelerometer measures, and the secondary objective was to investigate the factors associated with adherence to PA guidelines.

### Methods

Using data from the Korea National Health and Nutrition Examination Survey (KNHANES VII-2) 2017, we assessed 425 older adults who provided both self-reported and accelerometer data. Adherence to PA guidelines was evaluated using two different thresholds for the accelerometer data, each applied both strictly and with a tolerance of one to two minutes when defining a 10-minute bout.

### Results

Self-reported data indicated a 34% adherence rate, whereas accelerometer-based rates ranged from 16% to 62.8%, depending on the cutoff values and tolerance settings. Bland-Altman analyses demonstrated notable differences between subjective and objective measurements. In terms of correlates, the questionnaire

**Data availability statement:** The datasets analyzed for this study can be found in the seventh Korea National Health and Nutrition Examination Survey (KNHANES VII-2), 2017, Korea Disease Control and Prevention Agency (https://knhanes.kdca.go.kr/knhanes/eng/main. do under Survey Data → Data Introduction). The SAS code used for data preprocessing was excerpted from references [35,36], which are subject to the Korea Open Government License (KOGL) 4. While the code itself may be freely used upon proper citation of these references, the references themselves remain under the KOGL 4 license. The R code was authored by the present researchers and is made available without any restrictions.

**Funding:** This work was supported by the National Research Foundation of Korea (NRF) grant funded by the Korea government (MSIT) (No. 2022R1A5A8023404), awarded to C.H.L. The funders' website is available at https://www. nrf.re.kr. The funders had no role in the study design, data collection and analysis, decision to publish, or preparation of the manuscript.

**Competing interests:** The authors have declared that no competing interests exist.

**Abbreviations:** PA, physical activity; KNHANES, Korea National Health and Nutrition Examination Survey; CPM, counts per minute; GPAQ, Global Physical Activity Questionnaire; MVPA, moderate-to-vigorous physical activity; BMI, body mass index; HDL, high-density lipoprotein; OR, odds ratio; CI, confidence interval; LASSO, least absolute shrinkage and selection operator; SD, standard deviation; K-GPAQ, Korean Global Physical Activity Questionnaire.

data highlighted factors such as education and household income, whereas the accelerometer-based findings emphasized sex, age, hypertriglyceridemia, and low high-density lipoprotein cholesterol.

## Conclusions

This study revealed substantial discrepancies in both moderate-to-vigorous PA adherence estimates and significant predictors when comparing self-reported surveys to accelerometer data among older adults. For estimating population-level adherence, in the absence of accelerometer cutoff values established by a specialized lab, both self-reported data and accelerometer measurements offer unique insights. Meanwhile, when analyzing the factors influencing PA adherence, accelerometer data may be preferable, as subjective biases in self-report can affect the observed correlates in statistical results.

## Introduction

Physical activity (PA) has been well-established as beneficial for people of all ages, including older adults. These include prevention of metabolic and cardiovascular diseases [1,2] and reduction of mental problems [3,4]. Therefore, increasing PA levels has been recommended as public guidelines in many organizations and countries [5–10]. For instance, the World Health Organization [8] and the Korean Ministry of Health and Welfare [10] recommend that, for substantial health benefits, all adults aged 18 or over should perform at least 150–300 minutes of moderate-intensity aerobic PA or at least 75–150 minutes of vigorous-intensity aerobic PA, or an equivalent combination of moderate- and vigorous-intensity activity throughout the week.

Aerobic PA levels can be estimated using both subjective and objective measurements. Subjective methods using validated questionnaires, such as the International Physical Activity Questionnaire [11] and the Global Physical Activity Questionnaire (GPAQ) [12,13] are widely used, especially in studies, owing to ease of administration and affordability [14,15]. However, these subjective tools have inherent limitations in accurately capturing the full range of daily living activities and are prone to recall bias [15]. Objective methods using motion sensor technology, including accelerometers and pedometers have been occasionally used in National Health Surveys in several countries [16,17]. They can measure various type of activities, and even capture low-intensity activities objectively and accurately [15,18]. However, these technologies have limitations related to data reduction algorithms, particularly in the standardization of non-wear criteria and intensity cut-points, which may yield inconsistencies when evaluating PA across different studies [15,19,20]. Both methods have pros and cons; thus, they have been used in accordance to the users' objectives.

Previous studies have provided valuable insights into the complexities of PA assessment for older adults. For example, Shiroma et al. examined the correlation between self-reported and accelerometer-assessed moderate to vigorous physical activity (MVPA) among older women [21]. Their study, employing various cutoff points

including triaxial data, highlighted significant discrepancies in adherence rates based on the definition of MVPA used, thereby demonstrating the challenges in accurate PA assessment. Studies such as [22] and [23] have highlighted the variable relationship between subjective and objective measures of PA, suggesting that each may capture different facets of PA relevant to cardiovascular risk factors and daily activity levels, respectively. The work of [24] further elaborates on this complexity by additionally considering pedometer results and the construct validity of different PA measures and their associations with health outcomes. Furthermore, findings from [25] have shown the divergent results that can arise between self-reported and accelerometer-based estimates, particularly in relation to sedentary behavior.

In 2020, 45.6% of Korean adults met the aerobic PA guidelines based on questionnaire data, with several biological, demographic, social, and environmental factors influencing PA levels [26]. With metabolic disease prevalence rising with age, promoting PA in older adults is critical. Our research addresses one particular gap by examining PA adherence and its determinants among older Korean adults, a relatively underexplored area. Accelerometers, due to their sensitivity to even light and brief activity, are particularly useful within this demographic [27,28], whereas subjective PA data from older adults is often unreliable due to their memory impairments and misinterpretation [29,30].

There is no consensus on the optimal cutoff level for PA levels for older adults, and a wide range of cutoff levels has been used in previous studies to define MVPA, from 574 to 3250 counts per minute (cpm), when analyzing accelerometer-based data [31]. In this study, we incorporated both the commonly used cutoff level of >1952 cpm for MVPA in middle-aged adults [31–33] and the > 1040 cpm threshold derived from a laboratory-based calibration study in an older adult population [27]. In the latter study, the cutoff was established based on oxygen consumption—a representative health indicator of PA—and no significant sex differences were observed in either activity counts or oxygen consumption, thereby eliminating the need for sex-specific criteria. Meanwhile, because the GPAQ generally measures aerobic PA in bouts of at least 10 minutes, PA adherence—defined as achieving ≥150 minutes of MVPA within 7 days—was assessed by summing the total duration of continuous MVPA bouts of 10 minutes or more. In addition, we evaluated an alternative approach that allows a 1–2-minute tolerance when defining 10-minute bouts [34], leading to four scenarios: two cutoffs (1952 cpm and 1040 cpm), each applied with and without the 1–2-minute tolerance.

The primary objective of this study is to explore PA adherence among older adults in Korea, using both self-reported questionnaires and accelerometer measurements. The secondary objective is to investigate factors associated with this PA adherence based on both measurement tools.

## Materials and methods

### Sample

The Korea National Health and Nutrition Examination Survey (KNHANES) is an annual nationwide health survey conducted since 1998 by the Korea Disease Control and Prevention Agency. It aims to produce representative and reliable statistics on the health and nutritional status of the Korean population. This survey is publicly available annually and has been widely used in research to assess the epidemiology of chronic diseases, analyze health and nutritional status, examine socioeconomic disparities, and evaluate the effectiveness of health intervention in policy making.

The first dataset, HN17_ALL [35], collects data through physical examinations, measurements, and laboratory tests, as well as through health interviews covering sociodemographic, health-behavioral, and nutritional information, including self-reported PA levels measured using the Korean GPAQ (K-GPAQ) [13]. In the second dataset, HN17_PAM [36], objective PA levels were measured via accelerometers in a subset of 491 older adults (≥ 65 years) who were physically capable of mobility and consented to accelerometer-based PA measurement. This accelerometer dataset was provided as part of the KNHANES VII-2 2017. For our analysis, we integrated sociodemographic and health-related information from the full KNHANES dataset (HN17_ALL) with the accelerometer-based PA subset (HN17_PAM) using participants' IDs, allowing for a combined analysis of subjective and objective PA measurements.

Fig 1 presents the selection process used to derive our analytic sample from the whole dataset, illustrating how older adults (≥65 years) meeting the study's inclusion criteria were identified and further filtered based on accelerometer-based PA data availability. From the dataset (N = 8,127), a total of 6,456 individuals under 65 years of age were excluded, leaving 1,671 older adults (≥ 65 years). Of these, 1,180 did not consent to accelerometer-based PA measurement, resulting in 491 participants who were initially invited to wear an accelerometer. Among those invited, 17 were excluded due to device loss or malfunction, noncompliance, or data errors, and another 49 were excluded for having fewer than three valid days of accelerometer wear time. For participants who did not have 7 days of valid wear, the total minutes of MVPA were adjusted according to the number of valid days they had. Ultimately, we acquired a subset of PA monitoring data comprising 425 participants (197 men, 228 women) who had at least three valid days of accelerometer data from the KNHANES VII-2 2017 dataset, which was accessed for research purposes on February 15th 2023.

This project and all related materials were approved by the Pukyong National University Institutional Review Board (IRB: 1041386–202302-HR-17–02). No individual participants' identifiable data is included in this study and therefore informed consent for publication is not applicable.

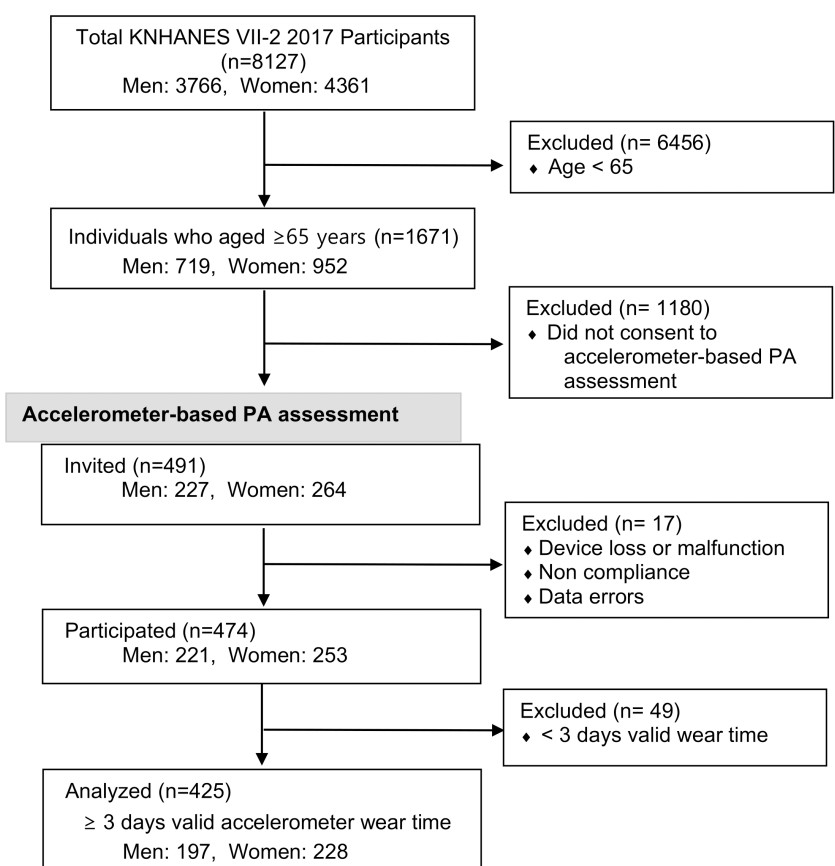

**Fig 1. Flow chart utilized to identify eligible male and female participants for the study.** KNHANES: Korea National Health and Nutritional Examination Survey, PA: physical activity.

## Measurement of physical activity

**Questionnaire data.** Questionnaire data were collected from participants using the K-GPAQ. These questionnaires gathered information about participants' continuous aerobic PA, which lasted for at least 10 minutes, including details about its intensity, frequency, and duration over three domains of daily PA: work, transportation, and leisure time. PA adherence was examined by calculating the total minutes of MVPA per week. This comprehensive approach allowed us to understand the context of the participants' aerobic PA in relation to their daily lives, including their adherence to PA guidelines.

To ensure the validity of the questionnaire methodology in the Korean context, the GPAQ was translated into Korean (referred to as K-GPAQ), and its validity was rigorously evaluated [13]. The K-GPAQ's reliability was assessed using a test-retest method, while its criterion-related validity was examined against accelerometer data. Although the K-GPAQ has shown a tendency to overestimate PA levels, it is considered reliable for capturing general trends in PA behaviors. This validation of the questionnaire method is essential for our study, as it ensures that the data collected provide a trustworthy representation of the participants' self-reported PA, complementing the objective data obtained from accelerometers.

**Accelerometer data.** Data were obtained from participants who agreed to wear an accelerometer (ActiGraph GT3X+, USA), which is a device used to assess PA levels. The accelerometers were affixed to the waist, aligned to either the left or right side of the navel, for 7 consecutive days, starting from the day after they agreed to participate. The device started collecting data at midnight on the consent day and was worn throughout the day, barring activities such as swimming or bathing, which could damage the device.

To analyze the collected data, we used SAS version 9.4 (SAS Institute Inc.) to identify and exclude non-wear time based on the data analysis guideline [36]. A wear time equal to or greater than 600 minutes per day was required for a valid wear day [34]. Because PA questionnaires may not strictly reflect 10-minute bouts of continuity as accurately as accelerometer measurements, we considered alternative definitions. One such proposed definition involves a 10-minute bout with an allowance for interruptions of up to 1–2 minutes [34]. This less stringent criterion recognizes sporadic disruptions that often occur in everyday life, and enables a more realistic and comprehensive understanding of an individual's engagement in MVPA. Consequently, four scenarios were considered using two cutoffs (1952 cpm and 1040 cpm), each assessed with and without the 1–2-minute tolerance.

**Physical activity adherence.** PA adherence was defined as engaging in 150–300 minutes of moderate-intensity aerobic PA or at least 75–150 minutes of vigorous-intensity aerobic PA, or an equivalent combination of moderate- and vigorous-intensity activity throughout the week, as measured using K-GPAQ. For the accelerometer-based assessment, we analyzed four distinct scenarios to capture MVPA bouts of at least 10 consecutive minutes, differentiated by cutoff values (1952 cpm or 1040 cpm) and the allowance of a brief tolerance period (1–2 minutes) within each bout. Specifically, the four scenarios are:

a) 1952 cpm 10-minute bout: Participants must maintain ≥1952 cpm for the full 10-minute period.

b) 1952 cpm 10-minute bout with a 1–2-minute tolerance: Participants must maintain ≥1952 cpm for at least 8 out of 10 minutes, permitting up to 2 minutes below the threshold.

c) 1040 cpm 10-minute bout: Participants must maintain ≥1040 cpm for the entire 10-minute interval.

d) 1040 cpm 10-minute bout with a 1–2-minute tolerance: Participants must maintain ≥1040 cpm for at least 8 out of 10 minutes, allowing up to 2 minutes below the threshold.

If the total MVPA minutes computed from the accelerometer reached or exceeded 150 minutes per week under any scenario, the participant was considered to have adhered to the PA guidelines.

## Sociodemographic and health-related variables

Several sociodemographic and health-related variables were considered in this study based on the KNHANES VII-2 2017 data. Sociodemographic variables included age, sex, education, household income, economic activity, and marital status, and health-related variables included smoking, alcohol consumption, body mass index (BMI), hypertension, diabetes, hypercholesterolemia, hypertriglyceridemia, and low high-density lipoprotein (HDL) cholesterol, unmet medical needs, depression, and activity limitation. These variables were carefully selected from among the extensive KNHANES dataset based on previous research (e.g., [22,33]) and preliminary statistical assessments to ensure both generalizability and analytical feasibility.

Regarding lifestyle factors, smoking status was classified into two groups: non-smokers, including those who have smoked less than 100 cigarettes in their lifetime, and smokers, who have smoked 100 or more cigarettes in their lifetime. Alcohol consumption was classified into two groups: non-drinkers, which included lifetime non-drinkers and those who consumed less than one drink per month over the past year, and drinkers, which included those who consumed one or more drinks per month over the past year.

BMI was calculated by dividing the weight in kilograms by the square of the height in meters (kg/m^2). Hypertension was defined as having a systolic blood pressure of 140 mmHg or higher, or a diastolic blood pressure of 90 mmHg or higher, or current antihypertensive drug use. Diabetes was identified when fasting blood glucose levels were ≥ 126 mg/dL, or if the individual had a clinical diagnosis of diabetes and was taking glucose-lowering medication, or receiving insulin injections. Hypercholesterolemia was defined as total cholesterol levels ≥ 240 mg/dL or current use of cholesterol-lowering drugs. Hypertriglyceridemia was defined as triglyceride levels ≥ 200 mg/dL. Low HDL cholesterol levels were defined as levels below 40 mg/dL for men and below 50 mg/dL for women.

Unmet medical needs were defined by the proportion of individuals who did not receive a medical examination or treatment despite needing it in the past year. Sustained depression was identified by the continuous presence of depressive feelings for at least 2 weeks. Activity limitation was defined as having any limitation in daily life or social activities due to health problems, physical disabilities, or mental disabilities.

## Statistical analysis

Statistical analysis was performed using SAS version 9.4 [37], R version 4.4.2 [38], and additional R packages. Specifically, the *dplyr* package was utilized for data preprocessing [39], the *car* package for assessing multicollinearity [40], the *mice* package for applying multiple imputation [41], the *glmnet* package for least absolute shrinkage and selection operator (LASSO) logistic regression analysis [42,43], and the *ggplot2* package for generating Bland-Altman plots [44]. Independent t-tests and chi-square tests were performed to compare continuous and categorical variables, respectively. When testing categorical variables, Fisher's exact test was used if the expected cell count was less than 5.

Logistic regression models were utilized to differentiate participants adhering to guidelines from those who did not. The analysis was conducted using three logistic regression models to assess the association between each characteristic. Model 1 included age sex, and activity limitations as baseline covariates and then assessed the effect of each additional covariate individually, thereby representing a minimal evaluation of each variable's independent association. Model 2 was fully adjusted, incorporating all 17 covariates simultaneously to evaluate their collective influence on PA adherence. Model 3 employed a two-step LASSO logistic regression approach: first, only first-order terms were considered, selecting three to five variables based on the LASSO path; second, all variables along with their interaction terms were entered into another LASSO analysis, allowing for the inclusion of one possible additional variable. The final model thus comprised those initially selected variables plus any significant interaction terms (and their main effects if not already included). This multi-model strategy enabled us to explore both minimal and comprehensive covariate adjustments, as well as to systematically identify potentially important interactions in a data-driven manner. In addition, sensitivity analyses were conducted, where the primary analyses were replicated using different criteria (i.e., scenarios a to d in the PA adherence subsection) for defining MVPA. All estimates were derived from multiple imputation datasets. A p-value of less than 0.05 was considered statistically significant.

Bland-Altman analysis was performed to determine whether subjective and objective measures of PA, for both minutes of MVPA per day, were in agreement. The following formulas were used to produce the Bland-Altman plots:

$$\text{Mean} = \frac{\text{K} - \text{GPAQ minutes of MPVA per day} + \text{accelerometer minutes of MVPA per day}}{2}, \quad (1)$$

$$\text{Difference} = \text{K} - \text{GPAQ minutes of MVPA per day} - \text{accelerometer minutes of MVPA per day}. \quad (2)$$

Here, Equation 1 was applied to the x-axis, and Equation 2 was applied to the y-axis. Additionally, the limits of agreement were calculated as follows:

$$\textit{Limits of agreement} = \mu_{Difference} \pm 1.96 \times \sigma_{Difference}, \quad (3)$$

where $\mu_{Difference}$ and $\sigma_{Difference}$ denote the mean and standard deviation (SD), respectively, of the difference defined in Equation 2.

## Results

### Demographic characteristics

Table 1 delineates the demographic characteristics of the participants who provided valid data by gender. The average age of the participants was 71.9±4.7 years, and ranged from 65 to 80 years. In contrast to women, men tended to have higher rates of alcohol consumption and cigarette smoking. In terms of health conditions, a larger proportion of women were diagnosed with hypercholesterolemia and low LDL cholesterol than were men.

Table 2 presents an exhaustive comparison of participant characteristics in relation to their adherence to the weekly 150-minute MVPA guidelines, as evaluated by the questionnaire and across four accelerometer scenarios—two cutoffs (>1040 cpm and >1952 cpm), each with and without a 1–2-minute tolerance. According to the questionnaire responses, 34% of participants (n = 144) met the guidelines, whereas 66% (n = 280) did not. In contrast, estimates based on accelerometer data varied markedly depending on the cutoff values used and whether a 1–2-minute tolerance was applied. For instance, using the higher cutoff of >1952 cpm without tolerance yielded an adherence rate of 16%, which nearly doubled (29.9%) when the tolerance was introduced. Meanwhile, applying the lower cutoff of >1040 cpm produced an adherence rate of 36.7%, rising to 62.8% with the addition of the tolerance. Overall, these findings highlight the sensitivity of accelerometer-based adherence rates to both the chosen cutoff threshold and the definitions of a bout.

From the test results in Table 2, it is possible to conduct a basic form of correlation analysis. For example, while no significant relationship was found between gender and adherence rates in the questionnaire data, a significant relationship was observed in the accelerometer data. For age, a significant relationship was identified across all datasets. Although these simple comparisons are useful for referencing overall patterns in adherence across variables, they are limited by the fact that they cannot account for multiple variables at once. Therefore, in the following subsection, we explore in greater depth which variables are correlated with adherence rates using more sophisticated models.

### Correlates of PA guidelines adherence

In this section, we address the correlates of PA guidelines adherence among older adults using three measurement approaches: (1) self-reported data from the K-GPAQ, (2) accelerometer data with a > 1952 cpm cutoff and a 1–2-minute tolerance, and (3) accelerometer data with a > 1040 cpm cutoff. Due to space limitations, results for the > 1952 cpm cutoff (no tolerance) and for the > 1040 cpm cutoff with tolerance are presented in S1 and S2 Tables in the Supporting Information.

**Table 1. Study participant demographics by gender and total (n = 425).**

| | Men | Women | Total | p-value |
|---|---|---|---|---|
| **N (%)** | 197 (46.4) | 228 (53.6) | 425 (100) | 0.133[a] |
| **Age, mean (SD)** | 72.4 (4.6) | 71.5 (4.7) | 71.9 (4.7) | **0.041[b]** |
| **Education** | | | | **<0.001[c]** |
|  Elementary school graduate or less, n (%) | 66 (33.5) | 136 (59.6) | 202 (47.5) | |
|  Middle school graduate, n (%) | 31 (15.7) | 36 (15.8) | 67 (15.8) | |
|  High school graduate, n (%) | 61 (31.0) | 41 (18.0) | 102 (24.0) | |
|  College graduate or higher, n (%) | 39 (19.8) | 15 (6.6) | 54 (12.7) | |
| **Standardized household income, mean (SD)** | 0.132 (1.03) | −0.114 (0.97) | 0.000 (1.00) | **0.012[b]** |
| **Economic activity** | | | | 0.190[c] |
|  With, n (%) | 76 (38.6) | 73 (32.0) | 149 (35.1) | |
|  Without, n (%) | 121 (61.4) | 155 (68.0) | 276 (64.9) | |
| **Marital status** | | | | **<0.001[c]** |
|  Married & living with partner, n (%) | 182 (92.4) | 123 (53.9) | 305 (71.8) | |
|  Otherwise, n (%) | 15 (7.6) | 105 (46.1) | 120 (28.2) | |
| **Lifetime smoking: 5 or more packs** | | | | **<0.001[c]** |
|  With, n (%) | 152 (77.2) | 7 (3.1) | 159 (37.4) | |
|  Without, n (%) | 45 (22.8) | 221 (96.9) | 266 (62.6) | |
| **Drinking alcohol** | | | | **<0.001[c]** |
|  With, n (%) | 122 (61.9) | 51 (22.4) | 173 (40.7) | |
|  Without, n (%) | 75 (38.1) | 177 (77.6) | 252 (59.3) | |
| **Body mass index, mean (SD)** | 23.7 (2.9) | 24.8 (3.2) | 24.3 (3.1) | **0.001[b]** |
| **Metabolic comorbidities** | | | | |
|  **Hypertension** | | | | 0.455[c] |
|   With, n (%) | 113 (57.4) | 140 (61.4) | 253 (59.5) | |
|   Without, n (%) | 84 (42.6) | 88 (38.6) | 172 (40.5) | |
|  **Diabetes** | | | | 1.000[c] |
|   With, n (%) | 45 (24.3) | 51 (24.3) | 96 (24.3) | |
|   Without, n (%) | 140 (75.7) | 159 (75.7) | 299 (75.7) | |
|  **Hypercholesterolemia** | | | | **0.003[c]** |
|   With, n (%) | 50 (27.0) | 88 (41.9) | 138 (34.9) | |
|   Without, n (%) | 135 (73.0) | 122 (58.1) | 257 (65.1) | |
|  **Hypertriglyceridemia** | | | | 0.767[c] |
|   With, n (%) | 19 (11.3) | 25 (12.9) | 44 (12.2) | |
|   Without, n (%) | 149 (88.7) | 169 (87.1) | 318 (87.8) | |
|  **Low HDL cholesterol** | | | | **<0.001[c]** |
|   With, n (%) | 49 (26.1) | 108 (49.8) | 157 (38.8) | |
|   Without, n (%) | 139 (73.9) | 109 (50.2) | 248 (61.2) | |
| **Unmet medical needs** | | | | **0.019[c]** |
|  With, n (%) | 11 (5.6) | 29 (12.7) | 40 (9.4) | |
|  Without, n (%) | 186 (94.4) | 199 (87.3) | 385 (90.6) | |
| **Sustained depression for more than 2 weeks** | | | | 0.027[c] |
|  With, n (%) | 19 (9.6) | 40 (17.5) | 59 (13.9) | |
|  Without, n (%) | 178 (90.4) | 188 (82.5) | 366 (86.1) | |
| **Activity limitations** | | | | 0.149[c] |
|  With, n (%) | 23 (11.7) | 39 (17.1) | 62 (14.6) | |
|  Without, n (%) | 174 (88.3) | 189 (82.9) | 363 (85.4) | |

*(Continued)*

**Table 1.** (Continued)

SD, standard deviation; HDL, high-density lipoprotein

[a]This p-value was obtained from a chi-square test assessing whether the overall gender distribution differed from a 1:1 ratio.

[b]Welch's t-tests were used to compare continuous variables.

[c]These categorical variables were compared using chi-square tests, while Fisher's exact tests were used if the expected cell frequency was < 5.

Table 3 presents the odds ratios (ORs) and associated 95% confidence intervals (CIs) for participant characteristics in relation to achieving ≥150 min/week of MVPA, as measured by the questionnaire. The analysis was conducted using three logistic regression models to assess the association between each characteristic, described in the Method section.

Certain participant characteristics were found to be significantly associated with achieving the recommended MVPA guidelines. In Model 1, higher education (OR=1.36 [1.12, 1.65]) and standardized household income (OR=1.33 [1.08, 1.65]) were each linked to higher odds of PA adherence, while older age (OR=0.95 [0.91, 1.00]) and activity limitations (OR=0.41 [0.19, 0.78]) emerged as negative correlates. In Model 2, the following variables were positively linked to meeting the PA guideline: being male (OR=2.37 [1.09, 5.24]), having higher educational attainment (OR=1.28 [1.03, 1.58]), and being diagnosed with hypercholesterolemia (OR=1.70 [1.05, 2.75]). Conversely, participants with activity limitations were significantly less likely to meet the guideline (OR=0.45 [0.20, 0.92]). Finally, Model 3, which was derived using a two-stage LASSO selection procedure, highlighted the following set of variables—education (OR=1.32 [1.08, 1.60]), standardized household income (OR=1.26 [1.02, 1.57]), and hypercholesterolemia (OR=1.73 [1.10, 2.71]) as positive correlates; hypertension (OR=0.60 [0.39, 0.93]) and activity limitations (OR=0.45 [0.21, 0.89]) as negative correlates. Although there were slight differences in the significant variables among the three models, they exhibited an overall similar pattern.

Table 4 provides factors associated with PA adherence measured by accelerometer using a threshold of 1952 cpm and bouts of at least 10 minutes with a tolerance of 1–2 minutes. Models 1 and 2 had the same set of significant variables, which are as follows—sex (being male) as a positive predictor (Model 1: OR=3.02 [1.91, 4.85] and Model 2: OR=3.94 [1.68, 9.50]); age (Model 1: OR=0.90 [0.85, 0.94] and Model 2: OR=0.89 [0.84, 0.95]), current economic activity (Model 1: OR=0.56 [0.34, 0.92] and Model 2: OR=0.58 [0.34, 0.97]), hypertriglyceridemia (Model 1: OR=0.24 [0.08, 0.58] and Model 2: OR=0.32 [0.10, 0.82]), and low HDL cholesterol (Model 1: OR=0.33 [0.19, 0.56] and Model 2: OR=0.37 [0.21, 0.64]) as negative predictors. Finally, Model 3 yielded a more parsimonious selection of significant predictors, retaining sex (OR=2.23 [1.40, 3.57]) while identifying two lipid-related factors—hypertriglyceridemia (OR=0.30 [0.10, 0.74]) and low HDL (OR=0.33 [0.19, 0.56])—as significant negative correlates. Although unmet medical needs and economic activity were selected during the LASSO variable selection process, they did not reach statistical significance in the final model constructed with the chosen variables. It is worthwhile noting that a non-significant result does not imply the absence of an effect; the corresponding OR values can remain meaningful.

Table 5 examines the OR for achieving the MVPA guidelines (>1040 cpm, without tolerance), as measured by an accelerometer. In Model 1, being male (OR=2.87 [1.89, 4.40]) was associated with higher odds of meeting the guidance, whereas older age (OR=0.92 [0.88, 0.96]), diabetes (OR=0.57 [0.34, 0.92]), hypertriglyceridemia (OR=0.50 [0.25, 0.97]), and low HDL cholesterol (OR=0.42 [0.26, 0.65]) had lower odds. In Model 2, males had higher odds (OR=3.07 [1.42, 6.82]), whereas older age (OR=0.93 [0.88, 0.99]) and low HDL cholesterol (OR=0.43 [0.26, 0.70]) continued to predict lower odds. Finally, in Model 3, male (OR=2.19 [1.44, 3.34]), diabetes (OR=0.60, [0.36, 0.97]), and low HDL (OR=0.41 [0.26, 0.64]) emerged as significant predictors.

## Assessment of measurement agreement

The subsequent section focuses on the agreement between the subjective and objective measurements of PA. The comparison is executed through the Bland-Altman method, a statistical approach designed to compare two measurement

Table 2. Participant characteristics based on adherence to 150 minutes per week MVPA guidelines measured by questionnaire and accelerometer.

| PA Adherence[a] | Questionnaire (K-GPAQ) | | Accelerometer (>1952 cpm) | | Accelerometer (>1952 cpm, tolerated) | | Accelerometer (>1040 cpm) | | Accelerometer (>1040 cpm, tolerated) | |
|---|---|---|---|---|---|---|---|---|---|---|
| | Met | Not met | Met | Not met | Met | Not met | Met | Not met | Met | Not met |
| N (%) | 144 (34.0) | 280 (66.0) | 68 (16.0) | 357 (84.0) | 133 (29.9) | 312 (70.1) | 156 (36.7) | 269 (63.3) | 267 (62.8) | 158 (37.2) |
| Sex \ p-value[b] | 0.154 | | 0.001 | | <0.001 | | <0.001 | | <0.001 | |
| Male, n (%) | 74 (51.4) | 122 (43.6) | 45 (66.2) | 152 (42.6) | 72 (63.7) | 125 (40.1) | 95 (60.9) | 102 (37.9) | 143 (53.6) | 54 (34.2) |
| Female, n (%) | 70 (48.6) | 158 (56.4) | 23 (33.8) | 205 (57.4) | 41 (36.3) | 187 (59.9) | 61 (39.1) | 167 (62.1) | 124 (46.4) | 104 (65.8) |
| Age \ p-value[c] | 0.030 | | 0.016 | | <0.001 | | 0.001 | | <0.001 | |
| Mean (SD) | 71.2 (4.4) | 72.3 (4.8) | 70.8 (3.9) | 72.1 (4.8) | 70.5 (4.0) | 72.4 (4.8) | 71.0 (4.3) | 72.5 (4.8) | 71.2 (4.5) | 73.2 (4.8) |
| Education \ p-value[b] | 0.001 | | 0.137 | | 0.050 | | 0.386 | | 0.090 | |
| Elementary school graduate or less, n (%) | 49 (34) | 152 (54.3) | 27 (39.7) | 175 (49) | 42 (31.6) | 160 (51.3) | 67 (42.9) | 135 (50.2) | 117 (43.8) | 85 (53.8) |
| Middle school graduate, n (%) | 26 (18.1) | 41 (14.6) | 9 (13.2) | 58 (16.2) | 19 (14.3) | 48 (15.4) | 24 (15.4) | 43 (16) | 49 (18.4) | 18 (11.4) |
| High school graduate, n (%) | 43 (29.9) | 59 (21.1) | 18 (26.5) | 84 (23.5) | 32 (24.1) | 70 (22.4) | 41 (26.3) | 61 (22.7) | 63 (23.6) | 39 (24.7) |
| College graduate or higher, n (%) | 26 (18.1) | 28 (10) | 14 (20.6) | 40 (11.2) | 20 (15) | 34 (10.9) | 24 (15.4) | 30 (11.2) | 38 (14.2) | 16 (10.1) |
| Standardized household income \ p-value[c] | 0.001 | | 0.080 | | 0.011 | | 0.058 | | 0.091 | |
| Mean (SD) | 0.246 (1.15) | −0.124 (0.89) | 0.201 (1.03) | −0.038 (0.99) | 0.212 (1.05) | −0.077 (0.97) | 0.121 (1.01) | −0.070 (0.99) | 0.060 (1.06) | −0.102 (0.88) |
| Economic activity \ p-value[b] | 0.505 | | 0.042 | | 0.344 | | 0.865 | | 0.303 | |
| With, n (%) | 47 (32.6) | 102 (36.4) | 16 (23.5) | 133 (37.3) | 35 (31.0) | 114 (36.5) | 56 (35.9) | 93 (34.6) | 99 (37.1) | 50 (31.6) |
| Without, n (%) | 97 (67.4) | 178 (63.6) | 52 (76.5) | 224 (62.7) | 78 (69.0) | 198 (63.5) | 100 (64.1) | 176 (65.4) | 168 (62.9) | 108 (68.4) |
| Marital status \ p-value[b] | 0.775 | | 0.094 | | 0.002 | | 0.001 | | <0.001 | |
| Married & living with partner, n (%) | 105 (72.9) | 199 (71.1) | 55 (80.9) | 250 (70) | 94 (83.2) | 211 (67.6) | 127 (81.4) | 178 (66.2) | 212 (79.4) | 93 (58.9) |
| Otherwise, n (%) | 39 (27.1) | 81 (28.9) | 13 (19.1) | 107 (30) | 19 (16.8) | 101 (32.4) | 29 (18.6) | 91 (33.8) | 55 (20.6) | 65 (41.1) |
| Lifetime smoking: 5 or more packs \ p-value[b] | 0.973 | | 0.028 | | 0.020 | | 0.006 | | 0.002 | |
| With, n (%) | 53 (36.8) | 105 (37.5) | 34 (50.0) | 125 (35.0) | 53 (46.9) | 106 (34.0) | 72 (46.2) | 87 (32.3) | 115 (43.1) | 44 (27.8) |
| Without, n (%) | 91 (63.2) | 175 (62.5) | 34 (50.0) | 232 (65.0) | 60 (53.1) | 206 (66.0) | 84 (53.8) | 182 (67.7) | 152 (56.9) | 114 (72.2) |
| Drinking alcohol \ p-value[b] | 0.567 | | 0.117 | | 0.094 | | 0.024 | | 0.110 | |
| With, n (%) | 62 (43.1) | 111 (39.6) | 34 (50) | 139 (38.9) | 54 (47.8) | 119 (38.1) | 75 (48.1) | 98 (36.4) | 117 (43.8) | 56 (35.4) |
| Without, n (%) | 82 (56.9) | 169 (60.4) | 34 (50) | 218 (61.1) | 59 (52.2) | 193 (61.9) | 81 (51.9) | 171 (63.6) | 150 (56.2) | 102 (64.6) |
| Body mass index \ p-value[c] | 0.399 | | 0.473 | | 0.143 | | 0.766 | | 0.179 | |
| Mean (SD) | 24.5 (2.9) | 24.2 (3.2) | 24.1 (2.9) | 24.3 (3.1) | 24.0 (2.7) | 24.4 (3.2) | 24.2 (2.8) | 24.3 (3.2) | 24.1 (2.8) | 24.6 (3.5) |
| Metabolic comorbidities | | | | | | | | | | |
| Hypertension \ p-value[b] | 0.021 | | 0.060 | | 0.029 | | 0.020 | | 0.019 | |
| With, n (%) | 74 (51.4) | 178 (63.6) | 33 (48.5) | 220 (61.6) | 57 (50.4) | 196 (62.8) | 81 (51.9) | 172 (63.9) | 147 (55.1) | 106 (67.1) |
| Without, n (%) | 70 (48.6) | 102 (36.4) | 35 (51.5) | 137 (38.4) | 56 (49.6) | 116 (37.2) | 75 (48.1) | 97 (36.1) | 120 (44.9) | 52 (32.9) |
| Diabetes \ p-value[b] | 0.182 | | 0.513 | | 0.130 | | 0.026 | | 0.006 | |
| With, n (%) | 27 (20.0) | 69 (26.6) | 13 (20.3) | 83 (25.1) | 20 (18.5) | 76 (26.5) | 27 (17.9) | 69 (28.3) | 50 (19.7) | 46 (32.6) |
| Without, n (%) | 108 (80.0) | 190 (73.4) | 51 (79.7) | 248 (74.9) | 88 (81.5) | 211 (73.5) | 124 (82.1) | 175 (71.7) | 204 (80.3) | 95 (67.4) |

| PA Adherence[a] | Questionnaire (K-GPAQ) | | Accelerometer (>1952 cpm) | | Accelerometer (>1952 cpm, tolerated) | | Accelerometer (>1040 cpm) | | Accelerometer (>1040 cpm, tolerated) | |
|---|---|---|---|---|---|---|---|---|---|---|
| | Met | Not met | Met | Not met | Met | Not met | Met | Not met | Met | Not met |
| **Hypercholesterolemia\ p-value[b]** | 0.246 | | 0.744 | | 1.000 | | 1.000 | | 0.785 | |
| With, n (%) | 53 (39.3) | 85 (32.8) | 24 (37.5) | 114 (34.4) | 38 (35.2) | 100 (34.8) | 53 (35.1) | 85 (34.8) | 87 (34.3) | 51 (36.2) |
| Without, n (%) | 82 (60.7) | 174 (67.2) | 40 (62.5) | 217 (65.6) | 70 (64.8) | 187 (65.2) | 98 (64.9) | 159 (65.2) | 167 (65.7) | 90 (63.8) |
| **Hypertriglyceridemia\ p-value[b]** | 0.557 | | 0.015 | | 0.03 | | 0.022 | | 0.010 | |
| With, n (%) | 13 (10.4) | 31 (13.1) | 1 (1.7) | 43 (14.1) | 3 (3.1) | 41 (15.5) | 9 (6.7) | 35 (15.4) | 20 (8.6) | 24 (18.5) |
| Without, n (%) | 112 (89.6) | 205 (86.9) | 57 (98.3) | 261 (85.9) | 94 (96.9) | 224 (84.5) | 126 (93.3) | 192 (84.6) | 212 (91.4) | 106 (81.5) |
| **Low HDL cholesterol\ p-value[b]** | 0.495 | | <0.001 | | <0.001 | | <0.001 | | <0.001 | |
| With, n (%) | 50 (36.0) | 106 (40.0) | 12 (18.2) | 145 (42.8) | 21 (18.9) | 136 (46.3) | 37 (24.0) | 120 (47.8) | 84 (32.2) | 73 (50.7) |
| Without, n (%) | 89 (64.0) | 159 (60.0) | 54 (81.8) | 194 (57.2) | 90 (81.1) | 158 (53.7) | 117 (76.0) | 131 (52.2) | 177 (67.8) | 71 (49.3) |
| **Unmet medical needs\ p-value[b]** | 0.279 | | 0.077 | | 0.053 | | 0.074 | | 0.001 | |
| With, n (%) | 10 (6.9) | 30 (10.7) | 2 (2.9) | 38 (10.6) | 5 (4.4) | 35 (11.2) | 9 (5.8) | 31 (11.5) | 15 (5.6) | 25 (15.8) |
| Without, n (%) | 134 (93.1) | 250 (89.3) | 66 (97.1) | 319 (89.4) | 108 (95.6) | 277 (88.8) | 147 (94.2) | 238 (88.5) | 252 (94.4) | 133 (84.2) |
| **Sustained depression for more than 2 weeks\ p-value[b]** | 0.179 | | 0.261 | | 0.312 | | 0.133 | | 0.301 | |
| With, n (%) | 15 (10.4) | 44 (15.7) | 6 (8.8) | 53 (14.8) | 12 (10.6) | 47 (15.1) | 16 (10.3) | 43 (16.0) | 33 (12.4) | 26 (16.5) |
| Without, n (%) | 129 (89.6) | 236 (84.3) | 62 (91.2) | 304 (85.2) | 101 (89.4) | 265 (84.9) | 140 (89.7) | 226 (84.0) | 234 (87.6) | 132 (83.5) |
| **Activity limitations\ p-value[b]** | 0.007 | | 0.595 | | 0.215 | | 0.720 | | 0.206 | |
| With, n (%) | 11 (7.6) | 50 (17.9) | 8 (11.8) | 54 (15.1) | 12 (10.6) | 50 (16.0) | 21 (13.5) | 41 (15.2) | 34 (12.7) | 28 (17.7) |
| Without, n (%) | 133 (92.4) | 230 (82.1) | 60 (88.2) | 303 (84.9) | 101 (89.4) | 262 (84.0) | 135 (86.5) | 228 (84.8) | 233 (87.3) | 130 (82.3) |

SD, standard deviation; HDL, high-density lipoprotein; cpm, counts per minute

[a]At least 150–300 minutes of moderate-intensity aerobic PA or 75–150 minutes of vigorous-intensity aerobic PA per week, or an equivalent combination of both.

[b]These categorical variables were compared using the chi-square tests, while Fisher's exact tests were used if the expected cell frequency was <5.

[c]Welch's t-tests were used to compare continuous variables.

**Table 3. The odds ratios [95% CI] of participant characteristics associated with achieving ≥150 minutes per week of MVPA guidelines measured by questionnaire.**

| | Model 1[a] | | Model 2[b] | | Model 3[c] | |
|---|---|---|---|---|---|---|
| | OR [95% CI] | p-value | OR [95% CI] | p-value | OR [95% CI] | p-value |
| **Sex, male** | 1.37 [0.91, 2.07] | 0.135 | 2.37 [1.09, 5.24] | **0.031** | | |
| **Age** | 0.95 [0.91, 1.00] | **0.040** | 0.97 [0.91, 1.02] | 0.194 | | |
| **Education** | 1.36 [1.12, 1.65] | **0.002** | 1.28 [1.03, 1.58] | **0.023** | 1.32 [1.08, 1.60] | **0.006** |
| **Standardized household income** | 1.33 [1.08, 1.65] | **0.008** | 1.23 [0.99, 1.55] | 0.067 | 1.26 [1.02, 1.57] | **0.032** |
| **Economic activity** | 0.73 [0.47, 1.14] | 0.170 | 0.80 [0.50, 1.28] | 0.359 | | |
| **Marital status** | 0.72 [0.42, 1.23] | 0.230 | 0.67 [0.38, 1.19] | 0.173 | | |
| **Lifetime smoking: 5 or more packs** | 0.53 [0.28, 1.01] | 0.054 | 0.54 [0.27, 1.05] | 0.071 | | |
| **Drinking alcohol** | 0.90 [0.57, 1.43] | 0.670 | 0.93 [0.57, 1.52] | 0.773 | | |
| **Body mass index** | 1.04 [0.97, 1.11] | 0.300 | 1.05 [0.97, 1.13] | 0.233 | | |
| **Hypertension** | 0.67 [0.44, 1.03] | 0.066 | 0.62 [0.39, 1.00] | 0.050 | 0.60 [0.39, 0.93] | **0.022** |
| **Diabetes** | 0.76 [0.46, 1.21] | 0.251 | 0.70 [0.41, 1.17] | 0.174 | | |
| **Hypercholesterolemia** | 1.66 [1.07, 2.57] | **0.022** | 1.70 [1.05, 2.75] | **0.030** | 1.73 [1.10, 2.71] | **0.017** |
| **Hypertriglyceridemia** | 0.91 [0.48, 1.68] | 0.775 | 1.10 [0.55, 2.14] | 0.789 | | |
| **Low HDL cholesterol** | 0.96 [0.62, 1.48] | 0.858 | 1.16 [0.72, 1.87] | 0.531 | | |
| **Unmet medical needs** | 0.78 [0.34, 1.63] | 0.520 | 0.86 [0.38, 1.85] | 0.715 | | |
| **Sustained depression for more than 2 weeks** | 0.87 [0.44, 1.66] | 0.673 | 0.85 [0.42, 1.70] | 0.658 | | |
| **Activity limitations** | 0.41 [0.19, 0.78] | **0.010** | 0.45 [0.20, 0.92] | **0.036** | 0.45 [0.21, 0.89] | **0.029** |

OR, odds ratio; CI, confidence interval; HDL, high-density lipoprotein

[a]Model 1 includes age, sex, and activity limitations as covariates, along with each variable in the column being added one at a time.

[b]Model 2 includes all the variables simultaneously to examine their combined effects on the outcome.

[c]Model 3 uses a two-step LASSO: first selecting main effects, then considering interaction terms if warranted.

techniques, with a focus on the level of agreement and potential bias between them. This approach not only enables us to compare mean differences but also to evaluate the consistency of the differences across the range of measurements.

Fig 2 displays Bland-Altman plots for daily MVPA minutes comparing the questionnaire (Q = K-GPAQ) and the accelerometer under four different scenarios: panel A uses a > 1952 cpm cutoff (A1952); panel B applies a > 1952 cpm cutoff with a 1–2-minute tolerance (A1952t); panel C uses a > 1040 cpm cutoff (A1040); and panel D applies a > 1040 cpm cutoff with tolerance (A1040t). The horizontal axis represents the mean of the two measures, as defined in Equation 1, while the vertical axis shows their difference, as specified in Equation 2. In each plot, the dashed central line represents the average difference (bias) across all participants, and the solid lines represent the 95% limits of agreement (± 1.96 SD), calculated using Equation 3. A narrower spread between the solid lines indicates closer agreement, whereas a wider spread suggests greater variability between methods.

Panels A and B—which use a higher cutoff (>1952 cpm)—show average differences of 13.81 and 6.72 minutes/day, respectively, with the questionnaire reporting higher MVPA times. In contrast, panels C and D—which use a lower cutoff (>1040 cpm)—exhibit average differences of 2.34 and -15.77 minutes/day, respectively. This implies that applying a higher cutoff criterion for determining 10-minute bouts in the accelerometer data results in a lower recorded MVPA time. Moreover, the mean difference is numerically lower when applying a 1–2 minute tolerance: panel B shows a lower difference than panel A, and panel D a lower difference than panel C. This suggests that even with the same cutoff, incorporating a 1–2 minute tolerance in bout determination tends to increase the recorded MVPA time, as if the cutoff were effectively lowered. Although one combination (panel C, > 1040 cpm) exhibited a relatively small overall bias, the triangular boundaries observed in the Bland-Altman plots indicate a pronounced discrepancy between the questionnaire and accelerometer

**Table 4. The odds ratios [95% CI] of participant characteristics associated with achieving ≥150 minutes per week of MVPA guidelines measured by accelerometer (>1952 cpm, ≥10 minute bouts with 1~2 minute tolerance).**

| | Model 1[a] | | Model 2[b] | | Model 3[c] | |
|---|---|---|---|---|---|---|
| | OR [95% CI] | p-value | OR [95% CI] | p-value | OR [95% CI] | p-value |
| **Sex, male** | 3.02 [1.91, 4.85] | **<0.001** | 3.94 [1.68, 9.50] | **0.002** | 2.23 [1.40, 3.57] | **0.001** |
| **Age** | 0.90 [0.85, 0.94] | **<0.001** | 0.89 [0.84, 0.95] | **<0.001** | | |
| **Education** | 1.11 [0.89, 1.36] | 0.351 | 0.96 [0.76, 1.22] | 0.745 | | |
| **Standardized household income** | 1.11 [0.89, 1.38] | 0.339 | 1.02 [0.81, 1.29] | 0.841 | | |
| **Economic activity** | 0.56 [0.34, 0.92] | **0.023** | 0.58 [0.34, 0.97] | **0.041** | 0.72 [0.43, 1.17] | 0.187 |
| **Marital status** | 1.14 [0.62, 2.17] | 0.675 | 1.13 [0.58, 2.27] | 0.725 | | |
| **Lifetime smoking: 5 or more packs** | 0.68 [0.35, 1.33] | 0.262 | 0.70 [0.34, 1.43] | 0.329 | | |
| **Drinking alcohol** | 0.83 [0.49, 1.37] | 0.464 | 0.77 [0.44, 1.34] | 0.358 | | |
| **Body mass index** | 0.96 [0.89, 1.04] | 0.302 | 0.98 [0.90, 1.07] | 0.663 | | |
| **Hypertension** | 0.75 [0.47, 1.20] | 0.229 | 0.83 [0.49, 1.39] | 0.474 | | |
| **Diabetes** | 0.69 [0.40, 1.18] | 0.186 | 0.85 [0.46, 1.53] | 0.593 | | |
| **Hypercholesterolemia** | 1.31 [0.80, 2.11] | 0.279 | 1.40 [0.81, 2.41] | 0.226 | | |
| **Hypertriglyceridemia** | 0.24 [0.08, 0.58] | **0.004** | 0.32 [0.10, 0.82] | **0.029** | 0.30 [0.10, 0.74] | **0.016** |
| **Low HDL cholesterol** | 0.33 [0.19, 0.56] | **<0.001** | 0.37 [0.21, 0.64] | **0.001** | 0.33 [0.19, 0.56] | **<0.001** |
| **Unmet medical needs** | 0.46 [0.15, 1.16] | 0.127 | 0.47 [0.15, 1.25] | 0.160 | 0.44 [0.14, 1.11] | 0.108 |
| **Sustained depression for more than 2 weeks** | 0.99 [0.46, 2.00] | 0.972 | 0.95 [0.43, 2.01] | 0.890 | | |
| **Activity limitations** | 0.79 [0.38, 1.55] | 0.516 | 0.80 [0.35, 1.72] | 0.578 | | |

OR, odds ratio; CI, confidence interval; HDL, high-density lipoprotein

[a]Model 1 includes age, sex, and activity limitations as covariates, along with each variable in the column being added one at a time.

[b]Model 2 includes all the variables simultaneously to examine their combined effects on the outcome.

[c]Model 3 uses a two-step LASSO: first selecting main effects, then considering interaction terms if warranted.

data. These triangular patterns represent cases where one method recorded 0 minutes/day of MVPA: points along the upper boundary correspond to instances where MVPA was reported in the questionnaire but not detected by the accelerometer, whereas points along the lower boundary indicate cases where MVPA was detected by the accelerometer but not reported in the questionnaire.

## Discussion

This study aimed to explore adherence to PA guidelines and its associating factors among the older population in South Korea through self-reported questionnaires and accelerometer measurements. This study included a moderately sized group of participants, with 425 of the initial 491 invitees providing usable data.

There have been studies on PA measurement using both subjective and objective methods in older adults with contradictory results; even according to national guidelines, the actual adherence rates can differ significantly depending on the measurement method. For example, subjective PA prevalence was much higher than objective PA in many studies [21,30,45,46], whereas the reverse was observed in others [47,48].

In this study, a noteworthy observation was the discrepancy in adherence rates when measured using self-reported questionnaires versus accelerometer readings. First, the PA adherence rate derived from accelerometer data was directly influenced by the cutoff values. Moreover, analysis of correlates revealed that each measurement method produced different significant covariates. Our analysis suggests that one of the main reasons for this discrepancy is the inherent subjectivity of self-report measures. Survey data, by their nature, introduce additional "noise" stemming from respondents'

**Table 5. The odds ratios [95% CI] of participant characteristics associated with achieving ≥150 minutes per week of MVPA guidelines measured by accelerometer (>1040 cpm, ≥10 minute bouts).**

| | Model 1[a] | | Model 2[b] | | Model 3[c] | |
|---|---|---|---|---|---|---|
| | OR [95% CI] | p-value | OR [95% CI] | p-value | OR [95% CI] | p-value |
| **Sex, male** | 2.87 [1.89, 4.40] | **<0.001** | 3.07 [1.42, 6.82] | **0.005** | 2.19 [1.44, 3.34] | **<0.001** |
| **Age** | 0.92 [0.88, 0.96] | **<0.001** | 0.93 [0.88, 0.99] | **0.013** | | |
| **Education** | 0.99 [0.82, 1.21] | 0.959 | 0.91 [0.73, 1.13] | 0.396 | | |
| **Standardized household income** | 1.05 [0.85, 1.29] | 0.662 | 0.97 [0.78, 1.22] | 0.819 | | |
| **Economic activity** | 0.84 [0.54, 1.30] | 0.434 | 0.81 [0.50, 1.29] | 0.376 | | |
| **Marital status** | 1.20 [0.70, 2.10] | 0.512 | 1.25 [0.69, 2.28] | 0.458 | | |
| **Lifetime smoking: 5 or more packs** | 0.75 [0.39, 1.42] | 0.387 | 0.78 [0.40, 1.51] | 0.465 | | |
| **Drinking alcohol** | 1.01 [0.64, 1.60] | 0.962 | 1.00 [0.61, 1.63] | 0.991 | | |
| **Body mass index** | 1.01 [0.94, 1.08] | 0.802 | 1.04 [0.97, 1.12] | 0.281 | | |
| **Hypertension** | 0.72 [0.47, 1.11] | 0.138 | 0.72 [0.45, 1.15] | 0.164 | | |
| **Diabetes** | 0.57 [0.34, 0.92] | **0.025** | 0.64 [0.37, 1.09] | 0.103 | 0.60 [0.36, 0.97] | **0.041** |
| **Hypercholesterolemia** | 1.28 [0.82, 1.99] | 0.273 | 1.35 [0.83, 2.20] | 0.233 | | |
| **Hypertriglyceridemia** | 0.50 [0.25, 0.97] | **0.047** | 0.69 [0.32, 1.39] | 0.308 | | |
| **Low HDL cholesterol** | 0.42 [0.26, 0.65] | **<0.001** | 0.43 [0.26, 0.70] | **0.001** | 0.41 [0.26, 0.64] | **<0.001** |
| **Unmet medical needs** | 0.57 [0.24, 1.23] | 0.167 | 0.58 [0.24, 1.29] | 0.197 | | |
| **Sustained depression for more than 2 weeks** | 0.75 [0.38, 1.44] | 0.401 | 0.72 [0.36, 1.41] | 0.349 | | |
| **Activity limitations** | 1.08 [0.59, 1.94] | 0.802 | 1.27 [0.65, 2.45] | 0.486 | | |

OR, odds ratio; CI, confidence interval; HDL, high-density lipoprotein

[a]Model 1 includes age, sex, and activity limitations as covariates, along with each variable in the column being added one at a time.

[b]Model 2 includes all the variables simultaneously to examine their combined effects on the outcome.

[c]Model 3 uses a two-step LASSO: first selecting main effects, then considering interaction terms if warranted.

subjective perceptions—often referred to as recall bias, social desirability bias, and similar phenomena. More specifically, some participants may overestimate their PA, while others underestimate it, thus creating additional variations that can obscure or weaken the true associations with objective outcomes. Accelerometer data can capture PA objectively, free from subjective noise, resulting in cleaner signals. For instance, in the accelerometer data, not only age and sex—considered significant factors for PA—but also various biochemical and clinical variables such as hypertriglyceridemia, low HDL cholesterol, and diabetes were identified as significant covariates (e.g., Tables 4 and 5). However, with the exception of hypercholesterolemia, these associations did not appear in the questionnaire data (Table 3), indicating that self-reported measures failed to detect such relationships.

Furthermore, we observed that education emerged as highly significant when using self-reported data in Table 3. This finding suggests that higher levels of education correlate with greater adherence to PA guidelines. However, when accelerometer data were used, this relationship not only lost its statistical significance but also showed a reversed trend, with the OR falling below 1 in most cases (e.g., Model 2 in Table 4 and Models 1–2 in Table 5). This discrepancy could possibly reflect differences in participants' tendencies to over- or under-report their PA. For example, the results in Table 2 of the study by Buszkiewicz et al. (2020) show that individuals with higher levels of education tend to report higher PA adherence on questionnaires—although they did not use the exact same GPAQ criteria—than accelerometer data indicate [46]. This subjectivity can be further supported by the observation that the activity limitation covariate, which is measured by questionnaire, also emerges as significant in the questionnaire data (Table 3) but not in the accelerometer data (Tables 4,5).

When estimating overall PA adherence rates across a large population, individual over-reporting and under-reporting can partially cancel each other out, allowing for a reasonably reliable approximation of adherence rate. However, in

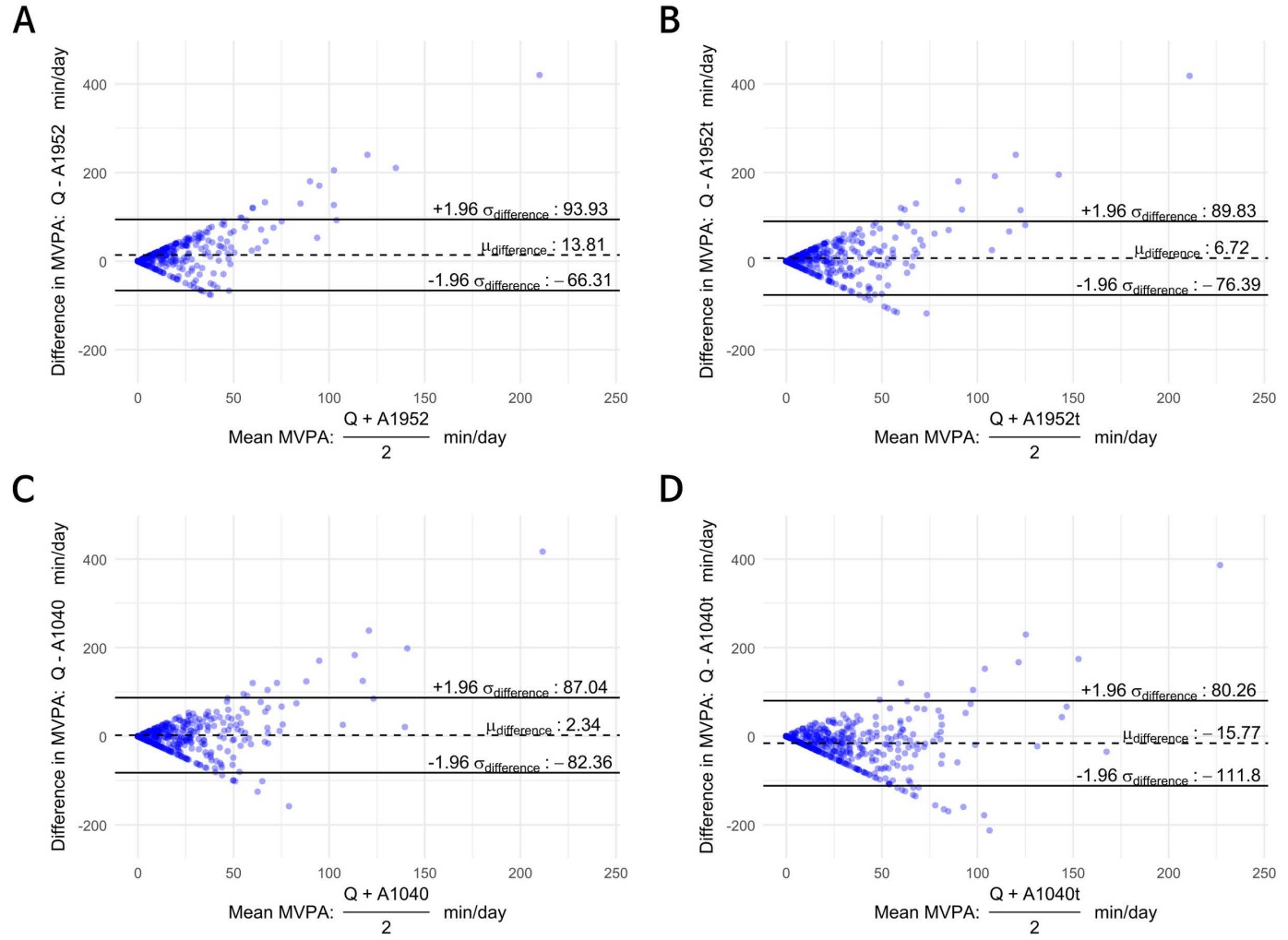

**Fig 2. Bland-Altman plots for daily MVPA minutes comparing the questionnaire data (Q = K-GPAQ) with accelerometer data under the following settings: A. A1952 (>1952 cpm). B. A1952t (>1952 cpm, tolerated). C. A1040 (>1040 cpm). D. A1040t (>1040 cpm, tolerated).**

correlational analyses—where noise in the outcome (i.e., PA adherence measure) directly impacts the significance of covariates—this subjective noise becomes more problematic. It can obscure genuine associations with objective variables, while simultaneously allowing certain covariates (such as education) to appear more significant because they capture the subjective bias itself. Consequently, from a correlational perspective, accelerometer data, which are free from the confounding influence of respondent subjectivity, tend to have an absolute advantage over questionnaire data.

Building on the perspective described above, we offer the following recommendations, which align with the two main objectives of this study. The first objective is to examine PA adherence among older adults in Korea by employing both self-report questionnaires and accelerometer measurements. For this objective, in cases where accelerometer cutoff values calibrated by a specialized lab are not available, we recommend drawing on both questionnaire-based and accelerometer-based data, rather than prioritizing one over the other. Although questionnaires inherently include subjective noise, the averaging process can mitigate some of its effects, and standardized questionnaire formats are recognized and used worldwide. By contrast, accelerometer data may introduce direct bias in PA adherence rates unless cutoff values appropriate to the population are applied. Therefore, in absence of a laboratory-calibrated cutoff threshold, accelerometer

data do not necessarily hold an absolute advantage over questionnaire data. We also do not endorse any specific accelerometer cutoff value simply because it yields, on average, results comparable to those obtained from questionnaire data.

On the other hand, for the second objective—investigating factors associated with PA adherence using both measurement tools—we recommend placing greater emphasis on accelerometer data. In our correlates analysis, we found that subjectivity issues directly affect the interpretation of the variables; therefore, accelerometer data offer an absolute advantage in this context.

Certain participant characteristics and adhering to PA guidelines were found to have significant associations, which should be considered when promoting PA in this age group. In particular, males were found to have higher odds of adhering to the guidelines than females, aligning with earlier research demonstrating higher PA levels in males [32,33]. The negative association between age and adherence to PA guidelines was also consistent with the existing literature [34], suggesting that as age increases, PA adherence tends to decrease. This might be a result of their comorbidities hindering PA, as well as socioeconomic and environmental barriers. Interestingly, health conditions, such as diabetes, hypertriglyceridemia, and low HDL cholesterol levels, were also found to be significant factors. It should be noted that the significant covariates obtained from observational studies only guarantee correlation rather than causation, so caution is warranted in their interpretation. One possible explanation or hypothesis is that non-adherence to PA guidelines may have led to these significant outcomes (i.e., diabetes, hypertriglyceridemia, and low HDL cholesterol). In other words, one could interpret that non-adherence has, in turn, influenced the covariates, resulting in the observed significant correlations. To infer causation, randomized controlled trials are essential.

In the logistic regression analysis, we explored a potential interaction using LASSO in Model 3. For the most part, the main findings did not reveal any significant interactions (Tables 3–5). However, when applying accelerometer data with a > 1040 cpm cutoff, ≥ 10-minute bouts, and a 1–2-minute tolerance (S2 Table), we detected significant interactions involving unmet medical needs and activity limitation. Specifically, although activity limitation on its own was not significant, it exerted a synergistic effect when combined with unmet medical needs, resulting in a substantial decrease in PA adherence. Given the large number of potential interaction terms tested, this finding may have arisen by chance (i.e., multiple testing). Hence, it should be interpreted with caution, and further research is required to confirm it.

Notably, the findings from the Bland-Altman plots demonstrate that two scenarios (i.e., b and c, which reflect a 1952 cpm 10-minute bout with a 1–2-minute tolerance and a 1040 cpm 10-minute bout, respectively) show minimal bias when comparing self-reported PA with objective measures. However, it should be noted that the questionnaire method is limited by respondents' subjectivity and therefore cannot serve as a standard for accelerometer cutoff values.

While this study provides important insights into the assessment of PA in older adults using both subjective and objective measures, it is imperative to acknowledge certain limitations that could influence the interpretation and generalizability of our conclusions. The primary limitations of our study are threefold. First, the KNHANES VII-2 dataset provided only the vertical axis component of the triaxial accelerometer data. The availability of all three axes could have allowed the application of alternative methodologies, such as vector-magnitude based algorithms, potentially offering a more comprehensive analysis of PA. Second, although the World Health Organization guidelines no longer apply this standard of a 10-minute bout, our study maintained this definition for consistency with the GPAQ. This approach may have overlooked the health benefits of shorter but intense physical activities. Third, the > 1040 cpm threshold used for determining MVPA was based on laboratory-based calibration studies from a population in a different country. Therefore, this threshold might not accurately reflect MVPA in the older population of South Korea. A more precise threshold for the Korean elderly population would ideally be determined through experiments conducted specifically within this demographic. Lastly, as PA levels are not necessarily stable over longer periods, individuals classified as more or less active at a single time point may not consistently remain so over time. Consequently, the cross-sectional nature of this study limits our ability to capture changes in PA behavior. Future research should consider longitudinal designs with objective PA measurements at multiple time points, ensuring consistent measures over time to more accurately capture temporal variations in activity levels.

## Conclusions

In conclusion, this study examined adherence to PA guidelines among older adults in Korea. We analyzed demographic characteristics, questionnaire responses, and accelerometer data according to different criteria. Furthermore, we specifically used Bland-Altman plots to compare and clarify the discrepancies between the questionnaire data and the accelerometer readings. Although our study compared accelerometer data with questionnaire responses, the main goal was not to prove the superiority of either method.

We observed significant demographic variations in PA guideline adherence, including sex, age, hypertriglyceridemia, and low HDL cholesterol levels. However, our study discovered discrepancies between questionnaire data and accelerometer data in analyzing the key factors associated with PA adherence, demonstrating that the potential subjectivity in questionnaire data may introduce issues in interpreting covariates.

From the standpoint of policy formulation, it is crucial to rely on objective evidence when assessing PA adherence and identifying its associated covariates. Based on our findings, in evaluating PA adherence rates among the target population (in this case, older adults in Korea), we recommend treating both questionnaire and accelerometer data as equally valid sources, since data reflecting diverse perspectives are essential for developing effective public policies. Meanwhile, we suggest prioritizing accelerometer data when exploring the correlates of PA adherence, given that the inherent subjectivity of questionnaire-based measures more directly affects their interpretation. Our findings provide a foundation for future research and interventions to promote PA adherence in this demographic, considering both subjective and objective aspects of PA measurement.

## Supporting information

**S1 Table. The odds ratios [95% CI] of participant characteristics associated with achieving ≥150 minutes per week of MVPA guidelines measured by accelerometer (>1952 cpm, ≥ 10 minute bouts).** OR, odds ratio; CI, confidence interval; HDL, high-density lipoprotein.
(DOCX)

**S2 Table. The odds ratios [95% CI] of participant characteristics associated with achieving ≥150 minutes per week of MVPA guidelines measured by accelerometer (>1040 cpm, ≥ 10 minute bouts with 1~2 minute tolerance).** OR, odds ratio; CI, confidence interval; HDL, high-density lipoprotein.
(DOCX)

**S1 File. Code and preprocessed data.** SAS code (freely reusable with proper citation of KOGL Type 4 references [35,36]), unrestricted R code authored by researchers, and preprocessed data are provided.
(ZIP)

## Acknowledgments

We sincerely thank the reviewers for their valuable comments and suggestions, which have greatly improved this manuscript. We would also like to thank Editage (www.editage.co.kr) for English language editing.

## Author contributions

**Conceptualization:** Su Hyun Kim, Young Hoon Kim.

**Data curation:** Junhui Park.

**Formal analysis:** Junhui Park.

**Funding acquisition:** Young Hoon Kim, Chang-Hyung Lee.

**Investigation:** Junhui Park, Su Hyun Kim.

**Methodology:** Junhui Park, Su Hyun Kim, Young Hoon Kim.

**Project administration:** Su Hyun Kim, Chang-Hyung Lee.

**Supervision:** Young Hoon Kim.

**Visualization:** Junhui Park.

**Writing – original draft:** Junhui Park.

**Writing – review & editing:** Su Hyun Kim, Young Hoon Kim, Chang-Hyung Lee.

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
