## [Decision Letter · Decision Letter 0]

28 Jan 2025

PONE-D-24-02389Assessing adherence to physical activity guidelines among older Korean adults with a focus on 10-minute bout duration using subjective and objective measuresPLOS ONE

Dear Dr. Kim,

Thank you for submitting your manuscript to PLOS ONE. After careful consideration, we feel that it has merit but does not fully meet PLOS ONE’s publication criteria as it currently stands. Therefore, we invite you to submit a revised version of the manuscript that addresses the points raised during the review process. The data is not technically sound good to effectively support the conclusions; the statistical analysis is not been rigorously performed appropriately and all the data are not underlying in the findings in this manuscript. So find below the reviewer remarks is important to take actions.  Please submit your revised manuscript by Mar 14 2025 11:59PM. If you will need more time than this to complete your revisions, please reply to this message or contact the journal office at plosone@plos.org. Please include the following items when submitting your revised manuscript:

We look forward to receiving your revised manuscript.

Kind regards,

Timoteo Salvador Lucas Daca, Ph.D

Academic Editor

PLOS ONE

“This work was supported by the National Research Foundation of Korea (NRF) grant funded by the Korea government (MSIT) (No. 2022R1A5A8023404).

This work was supported by the Pukyong National University Research Fund in 2022 (No. 202212470001).”

Reviewers' comments:

Reviewer's Responses to Questions

**Comments to the Author**

1. Is the manuscript technically sound, and do the data support the conclusions?

Reviewer #1: Yes

Reviewer #2: Partly

2. Has the statistical analysis been performed appropriately and rigorously? 

Reviewer #1: Yes

Reviewer #2: No

3. Have the authors made all data underlying the findings in their manuscript fully available?

Reviewer #1: No

Reviewer #2: Yes

4. Is the manuscript presented in an intelligible fashion and written in standard English?

Reviewer #1: Yes

Reviewer #2: Yes

5. Review Comments to the Author

Reviewer #1: GENERAL REMARKS

In the work entitled "Assessing adherence to physical activity guidelines among older Korean adults with a focus on 10-minute bout duration using subjective and objective measures" the authors sought to:

(i) analyze physical activity (PA) levels using both self-reported (via the validated K-GPAQ, Korea - Global Physical Activity Questionnaire) and objective accelerometer measures, as well as

(ii) investigate the factors associated with adherence to PA guidelines (using two different thresholds for the accelerometer data).

For this, the authors used data from the "Korea National Health and Nutrition Examination Survey (KNHANES VII-2) 2017" (aimed at individuals aged 65 years or older, who were capable of mobility and had consented to have their PA level measured by an accelerometer), in a study population consisting of n=363 older adults (average age 71.8 ± 4.74 yrs), across different socio-demographic characteristics. The authors conclude that: (i) discrepancies between self-reported and objective measurements were observed (more discrepant for the higher and lower cut-off values), and (ii) factors such as sex, age, and certain health conditions were significantly associated with PA adherence.

The study has its value. I will make some observations which can be of pertinence for the discussion regarding the study's more relevant inherent limitations and stronger points:

1) PA is not stable over longer periods which means that individuals identified as more or less active (or inactive) might not be consistently so; thus, it would be useful to perform longitudinal studies using objective measurements of PA, both at baseline and follow-up, with consistent measures over time;

2) The work https://doi.org/10.1016/j.bbr.2020.113061, by the same authors as REF 20, identified that >72% of studies evaluated PA "with self-report questionnaires, with limited validity in capturing the full range of daily living activities and with limitations associated with an inaccurate recall", while "in contrast, wearable sensors can provide an objective and precise assessment of everyday activities and may allow the identification of a specific PA pattern across time" in that "sensors incorporate measures of several activities including exercise, non-exercise, lifestyle activities (e.g., walking), and instrumental activities of daily life (e.g., housework). In general, these measures indicate the amount of total movement within an individual’s environment", concluding that "objective measures are extremely important to capture the low-intensity PA (particularly if considering that many older adults are sedentary)"

These aspects seem to be congruent with the authors observations (this reviewer does understand that the data used precludes the consideration of other measures across time), not taking away from the study, but enriching the body of literature.

ABSTRACT AND INTRODUCTION

Well written and aligned with the work. Given the existing body of literature, I would suggest that the introduction would benefit from including references to a couple of systematic reviews and/or meta-analysis on the matter, rather than limiting to individual studies (regardless of their merit). My suggestion derives from the fact that literature and guidelines seem to be rather US-centric (see your suggested REFS 5-8), and perhaps larger studies could provided for more generalizable/over-reaching conclusions. I realize that if those are based on US-centric studies/guidelines that there is also an argument here against it. I will, thus, leave this up to the authors.

I do, however, wonder: are there recommendations in Korea by, for instance, a health ministry (or equivalent government/health authority)? My question is not such a rethorical one - the paper proposes to assess "adherence to physical activity guidelines" - thus, it begs the question: whose (which) guidelines?

Of note, this objective is re-stated somewhat differently in lines 87 and 88, in which it is stated that the study aim is to "explore PA levels among older adults in Korea, using both self-reported questionnaires and accelerometer measurements".

I would make sure to align title, with objectives stated in the abstract, with objectives stated in the concluding paragraph of the introduction, and clearly state the guidelines using to determine/assess Adherence/Nonadherence.

MATERIALS AND METHODS

- In line 99, a "which" is required after the comma so to read ", which was accessed for research purposes on February 15th 2023". (note: I will not further dwell on minor grammatical lapses, as certainly a careful reading by the authors will identify those in latter phases);

- Are there other studies using the KNHANES VII-2 2017 dataset? Can references be provided? The KNHANES VII-2 2017 was coordinated by whom? It covered what percentage of the population? Was it country wide? Some general descriptives of the KNHANES VII-2 2017 are needed. I am also confused on this supplemented other self-reported PA? Was it KNHANES VII-2 2017 run after another survey, adding to the data? Lines 98 to 104 are essential in understanding the dataset at hand;

- Please better explain what is considered to be "capable of mobility" - there are different levels of mobility. This variable in itself may be of great consideration - in line 169 when you explain your models, I am tempted to argue that Model 1 should account for heterogeneity in mobility, and only then run other Models (including age and biological sex);

- Please provide the reasoning for the cpm values (>560, >1040, >1952) in conjunction with a 10-minute bout criterion. Did you follow other studies (for instance, REF 16 that you discuss in your intro)? Did you follow some pre data analysis that indicated for this cut-offs/tresholds? That is: where do these specific numbers came from?

- Did you use the GPAQ or the K-GPAQ? (please, compare and contrast information on line 123 and 130;

- Why the variables on lines 139 to 1443? Why were these specific ones selected from the KNHANES VII-2 2017 information source? Again, was it based on literature? Did you run some statistical analysis and these were the most significant? It is ok to show data for the most significant, but it would be great to also understand if there were others that were run but without significant results (in itself a finding), or the reasoning behind "picking" some and not others (albeit I do not think that being "significant" was your reasoning). The statement in line 159 a 160 does not withstand the test of scientific rigor on this (please provide REFS or a more sustainable/sound argument);

RESULTS

- On "Table 2" for the nomenclature/line name for the parameter "Physical activity adherence - Questionnaire" can be conducive to misinterpretation. For instance, this reviewer read the table in that not all participants have all measures (in this case that on=230 did not have data), while it means that 63.4% (n = 230) did not meet the guidelines (as explained in the paragraph). I would rename the line, or add a footnote, so that it is clearer.

CONCLUSION/DISCUSSION

- In the opening paragraph, the authors have caused some confusion on this reviewer. I was operating under the assumption (and the manuscript led me to it) that the authors used a "subset" of data of the larger survey KNHANES VII-2 2017. In fact, in the results lines 186 to 192, and Figure 1, I continued to read under this assumption, stemming from the materials & methods. However, in lines 307 to 308, now it clearly states that the study authors recruited the participants. I must stress the need to please better describe these samples: KNHANES VII-2 2017 vs. study sample - it has to be very clear.

- Lines 310 and 311, please insert something that states "even according to national Guidelines" after the ";" to read "results; XXXXX, subjective PA prevalence ...", so that the reader can grasp that are not just 1 study against another, but rather national guidelines/surveys/whatnot. Are 2010 and 2011 the most recent that indicate for this?;

- Why would, from your standpoint as researchers, and on line 321, more clinical/biochemical variables better associate with accelerometer (objective) measures? In the same manner, in line 325 to 332, you state findings, but seems to me that some discussion is needed: "These observations indicate that accelerometer-based, objectively measured PA, might reflect a different behavioral landscape than that reflected by selfreported measures". Which landscape? And why? Provide some food for thought!;

- Not really sure what the authors mean by the statement on line 352 "Moreover, the corresponding threshold setting is applicable to the older population in Korea". Applicable how? Is there a particular characteristic that makes it so? Previous data? Literature pointing in that direction? (this does back to my previous comment on why these tresholds).

Thank you for allowing me to review your work and looking forward for your reply!

Reviewer #2: This study aims to assess the adherence to PA guidelines in older Korean adults according to sociodemographic, lifestyle and health factors. PA was assessed by using a questionnaire and accelerometer recordings.

Review questions

Is the manuscript technically sound, and do the data support the conclusions?

Partly. Please see some comments.

Has the statistical analysis been performed appropriately and rigorously?

No. The statistical analyses are partly well performed. To complete the analyses, you could consider a model including interactions with a factor selection processing. Please consider the comments hereafter.

Have the authors made all data underlying the findings in their manuscript fully available?

Yes. It is written that this is available but the link provided leads to a blank page.

Is the manuscript presented in an intelligible fashion and written in standard English?

Yes. However, there are typos and some sentences should be revised.

Review Comments to the Author

General Comments

The study brings interesting information about the older Korean adult population, especially regarding PA habits through a study of factors associated with PA engagement.

However, some parts of the purpose of this study could be clearer (see details after). The results could be improved by using similar methods along the manuscript. Indeed, in the first part, there are comparisons between questionnaire, accelerometer > 1040 cpm and accelerometer > 1952 cpm whereas in the second part (Bland-Altman plots), the authors compared 1) questionnaire vs accelerometer > 1040 cpm and 2) accelerometer > 1952 cpm with 1 to 2 min tolerance. You could complete by adding information about “accelerometer > 1952 cpm with 1 to 2 min tolerance” in the first part and adding a Bland-Altman plot with “accelerometer > 1952 cpm”.

More specifically, the statistical analyses could be refined (see details after).

The discussion could be improved. It is not clear what the authors could recommend about the method to use regarding the threshold. In addition, discussing the health factors associated with adherence could add value to the study.

Specific Comments

Abstract

Adherence to PA guidelines was evaluated using two different thresholds for the accelerometer data.

In the results, there are 3 thresholds (1952, 1040 and 560).

Financial disclosure

If possible, enter the initials of the authors who received the awards and the URL of the funders’ website. Please, add the statement informing if the funders played any role in the study.

Data availability

The URL provided to access the data leads to a blank page (accessed on 22nd January 2025). Please, update the link.

Introduction

Line 88 – 89: I may miss something but I don’t understand why you would like to get a more accurate alignment with questionnaire data as it was a gold-standard fir PA assessment. When reading the paper, it seems you compared accelerometer assessment with different parameters and you found that one is more aligned than the other one. Please rephrase.

Statistical analysis

Line 113: results related to 560 are not showed in the document.

Line 117 – 121: Why didn’t you consider to use the “less stringent criterion” in the descriptive results and in the logistic models?

Line 162, 163: Please add a R citation in the text as recommended by the software providers.

Line 165: Make sure / check the use of t-tests is valid in each case. It may have been done since you used Welch t-tests in Table 2 (line 220).

Line 170: It seems you didn’t consider the interaction factors in Model 2, especially age vs other factors, sex vs other factors, smoking vs hypertension, etc. You should add the significant ones in Models 2 and 3 (see a note hereafter) but the number of factors will increase (cf note below).

Line 172: “the primary analyses were replicated using different thresholds”. Only one threshold is written. Please specify the thresholds you used.

Line 179: “accelrometer” -> “accelerometer” (typo).

Results

Line 186: “the study’s participation process” (to be rephrased).

Line 208 and Table 2: Why didn’t you consider the 560 cpm threshold? This threshold should probably be removed from the abstract and the methods.

Line 229 (Table 3), Line 257 (Table 4), Line 272 (Table 5): Model 2 involves many factors. The conditions for applying logistic regression may not be met since there only 133 participants with adherence (for Table 3). Suggestion: You might consider a Model 3 involving only factors with a p value less than a threshold (p < 0.2 for instance), such as “Drinking alcohol” (in Table 3). However, I don’t think your conclusions will be changed.

Table 5 vs Fig. 2: The result message may be clearer if the criteria are similar, e.g. “1952 cpm, bouts lasting ≥ 10 min” or “1952 cpm, bouts lasting ≥ 10 min with 1 to 2 min tolerance”.

Lines 285 – 287: You could help the reader to detect what brings the graph in your study.

Moreover, your manuscript includes only two graphs. What is the evidence showing that this threshold is optimal for minimizing the bias?

Discussion

Lines 325 – 332: What are your recommendations to readers? Should we consider that the associated factors are those highlighted with accelerometers, those highlighted with questionnaires, both? This is important to identify the factors on which public policy can focus on to address the lack of PA (cf Line 95).

Lines 339 – 341: Could you provide explanations or hypotheses regarding the association with these health factors?

Lines 342, 343: As highlighted in a comment about the introduction content, we could understand that using questionnaires is a gold-standard. Please rephrase.

Line 350 – 352: What evidence is there to support this assertion?

Line 350 – 358: What are your recommendations to readers? Which method is the best (threshold, with without tolerance)? How your results can support it?

Line 358: This may partly explain the discrepancies observed but did you get the information about the participant engagement in water-based physical activities?

Conclusion

Line 380, 381: What conditions enabled to get a minimal bias? This sentence is probably not appropriate since you did not compare biases according to factors but you compared the alignment according to algorithm parameters (thresholds) to detect MVPA. This should be rephrased.

Figures

Figure 2 and Figure 3: Please specify the units on the axes.

I find the shapes of the plots are a bit strange because of the linear pattern (like a triangle). Is there an explanation about these shapes?

6. PLOS authors have the option to publish the peer review history of their article (what does this mean? ). If published, this will include your full peer review and any attached files.

**Do you want your identity to be public for this peer review?** For information about this choice, including consent withdrawal, please see our Privacy Policy .

Reviewer #1: **Yes: ** Nadine Correia Santos

Reviewer #2: No

---

## [Author Response · Author response to Decision Letter 1]

17 Mar 2025

Thank you for the opportunity to submit the revised version of our manuscript. We sincerely appreciate the time and effort that you and the reviewers have invested in providing valuable feedback. We are grateful for the reviewers' insightful comments, which have helped us improve our work. In response, we have incorporated most of their suggestions and have uploaded a separate file detailing the revisions.

---

## [Decision Letter · Decision Letter 1]

24 Apr 2025

Assessing adherence to physical activity guidelines and correlates among older Korean adults with a focus on 10-minute bout duration using subjective and objective measures

PONE-D-24-02389R1

Dear Dr. Su Hyun Kim,

We’re pleased to inform you that your manuscript has been judged scientifically suitable for publication and will be formally accepted for publication once it meets all outstanding technical requirements.

Kind regards,

Timoteo Salvador Lucas Daca, Ph.D

Academic Editor

PLOS ONE

Additional Editor Comments (optional):

Reviewers' comments:

Reviewer's Responses to Questions

**Comments to the Author**

1. If the authors have adequately addressed your comments raised in a previous round of review and you feel that this manuscript is now acceptable for publication, you may indicate that here to bypass the “Comments to the Author” section, enter your conflict of interest statement in the “Confidential to Editor” section, and submit your "Accept" recommendation.

Reviewer #2: All comments have been addressed

2. Is the manuscript technically sound, and do the data support the conclusions?

Reviewer #2: Yes

3. Has the statistical analysis been performed appropriately and rigorously? 

Reviewer #2: Yes

4. Have the authors made all data underlying the findings in their manuscript fully available?

Reviewer #2: Yes

5. Is the manuscript presented in an intelligible fashion and written in standard English?

Reviewer #2: Yes

6. Review Comments to the Author

Reviewer #2: Dear authors,

In this new version, all my comments have been addressed.

I now only have minor comments hereafter.

Lines 59, 100, 101, 112: “studes”, “represenatativ”, “publicily”, “participatn” (typo)

Table 1: I didn’t notice this issue in the first round, but I don’t understand to what the first p-value in the table refers to. As far as I understand, the p-values shown are used to compare groups of men and women. But in this context, the first p-value (0.133) doesn’t make sense.

Table 3 to 5: Specify in the footnote (c) the interactions included in the model (and is possible their associated p-value).

line 460: "In general, it is relatively uncommon to observe an interaction effect." -> I think this is "uncommon" because the interaction effect is (sadly) rarely considered in the models. If you agree, you can rephrase or remove this sentence.

lines 467 - 469: The sentence is similar to the previous one.

Best luck for your submission.

Kind regards.

7. PLOS authors have the option to publish the peer review history of their article (what does this mean? ). If published, this will include your full peer review and any attached files.

**Do you want your identity to be public for this peer review?** For information about this choice, including consent withdrawal, please see our Privacy Policy .

Reviewer #2: No

---

## [Editor Report · Acceptance letter]

PONE-D-24-02389R1

PLOS ONE

Dear Dr. Kim,

I'm pleased to inform you that your manuscript has been deemed suitable for publication in PLOS ONE. Congratulations! Your manuscript is now being handed over to our production team.

Kind regards,

on behalf of

Dr. Timoteo Salvador Lucas Daca

Academic Editor

PLOS ONE